# Mystery(n) Phenotypic Presentation in Europeans: Report of Three Further Novel Missense *RNF213* Variants Leading to Severe Syndromic Forms of Moyamoya Angiopathy and Literature Review

**DOI:** 10.3390/ijms23168952

**Published:** 2022-08-11

**Authors:** Claudia Santoro, Giuseppe Mirone, Mariateresa Zanobio, Giusy Ranucci, Alessandra D’Amico, Domenico Cicala, Maria Iascone, Pia Bernardo, Vincenzo Piccolo, Andrea Ronchi, Giuseppe Limongelli, Marco Carotenuto, Vincenzo Nigro, Giuseppe Cinalli, Giulio Piluso

**Affiliations:** 1Child and Adolescent Neuropsychiatry Clinic, Department of Mental and Physical Health and Preventive Medicine, University of Campania “Luigi Vanvitelli”, Via Pansini 5, 80131 Naples, Italy; 2Department of Women’s and Children’s Health, and General and Specialized Surgery, University of Campania “Luigi Vanvitelli”, Via De Crecchio 4, 80138 Naples, Italy; 3Department of Neurosciences, Santobono-Pausilipon Children’s Hospital, AORN, Via Ravaschieri 8, 80122 Naples, Italy; 4Department of Precision Medicine, University of Campania “Luigi Vanvitelli”, Via Luigi De Crecchio 7, 80138 Naples, Italy; 5Department of Pediatrics, Santobono-Pausilipon Children’s Hospital, AORN, Via Ravaschieri 8, 80122 Naples, Italy; 6Department of Radiology, Tortorella Private Hospital, Via Aversano 1, 84214 Salerno, Italy; 7Laboratory of Medical Genetics, ASST Papa Giovanni XXIII, Piazza OMS 1, 24127 Bergamo, Italy; 8Department of Neurosciences, Pediatric Psychiatry and Neurology, Santobono-Pausilipon Children’s Hospital, AORN, Via Ravaschieri 8, 80122 Naples, Italy; 9Dermatology Unit, University of Campania “Luigi Vanvitelli”, Via Pansini 5, 80131 Naples, Italy; 10Anatomic Pathology Unit, University of Campania “Luigi Vanvitelli”, Piazza Miraglia 2, 80138 Naples, Italy; 11Division of Cardiology, Monaldi Hospital, University of Campania “Luigi Vanvitelli”, Via Bianchi, 80131 Naples, Italy; 12Telethon Institute of Genetics and Medicine, Via Campi Flegrei 34, 80078 Pozzuoli, Italy

**Keywords:** *RNF213*, moyamoya, whole exome sequencing, hypertransaminasemia, livedo reticularis, annular figurate erythema, skin, hypertension

## Abstract

Moyamoya angiopathy (MMA) is a rare cerebral vasculopathy in some cases occurring in children. Incidence is higher in East Asia, where the heterozygous p.Arg4810Lys variant in *RNF213* (Mysterin) represents the major susceptibility factor. Rare variants in *RNF213* have also been found in European MMA patients with incomplete penetrance and are today a recognized susceptibility factor for other cardiovascular disorders, from extracerebral artery stenosis to hypertension. By whole exome sequencing, we identified three rare and previously unreported missense variants of *RNF213* in three children with early onset of bilateral MMA, and subsequently extended clinical and radiological investigations to their carrier relatives. Substitutions all involved highly conserved residues clustered in the C-terminal region of RNF213, mainly in the E3 ligase domain. Probands showed a *de novo* occurring variant, p.Phe4120Leu (family A), a maternally inherited heterozygous variant, p.Ser4118Cys (family B), and a novel heterozygous variant, p.Glu4867Lys, inherited from the mother, in whom it occurred *de novo* (family C). Patients from families A and C experienced transient hypertransaminasemia and stenosis of extracerebral arteries. Bilateral MMA was present in the proband’s carrier grandfather from family B. The proband from family C and her carrier mother both exhibited annular figurate erythema. Our data confirm that rare heterozygous variants in *RNF213* cause MMA in Europeans as well as in East Asian populations, suggesting that substitutions close to positions 4118–4122 and 4867 of *RNF213* could lead to a syndromic form of MMA showing elevated aminotransferases and extracerebral vascular involvement, with the possible association of peculiar skin manifestations.

## 1. Introduction

Moyamoya angiopathy (MMA) is a rare cerebral vasculopathy in some cases occurring in children. It is characterized by progressive stenosis of the distal intracranial internal carotid artery (ICA) that also involves the proximal anterior cerebral arteries (ACAs) and middle cerebral arteries (MCAs), as well as the development of a hazy compensatory network of collateral arteries called moyamoya vessels [1,2,3,4].

MMA incidence is higher in East Asian countries, especially Korea, Japan, and China [5,6,7]. This is very likely due to genetic susceptibility factors in these populations [8,9]. *RNF213* (Mysterin) is the major susceptibility gene for MMA, with the heterozygous p.Arg4810Lys variant representing the founder mutation present in 70–90% of Japanese and Korean MMA patients [8,10,11,12]. Other variants of *RNF213* were recently found in European and in East and South Asian patients affected by MMA [10,13,14,15].

Although MMA is being increasingly reported among Europeans, very little is currently known about the penetrance, mode of inheritance, and clinical phenotype of patients with MMA and rare variants in *RNF213*. In these individuals, mainly missense variants preferentially clustered in the C-terminal region are associated with the risk of developing severe forms of the disease [16,17,18]. Differently to East Asian countries, no major founder variant was identified in European patients with MMA [19].

By whole exome sequencing, we genetically characterized three European families in which probands presented severe bilateral MMA associated with rare missense variants in *RNF213*. Our findings support the role of *RNF213* as a pathogenetic cause, and not merely a susceptibility factor, of MMA in Europeans. We also discuss the variability of clinical and neuroradiological phenotypes related to the identified variants in *RNF213* in Europeans based on a review of the existing literature.

## 2. Results

Probands and, when necessary, their carrier relatives underwent a comprehensive clinical and neuroradiological evaluation. Demographical, clinical, and radiological features of our patients are summarized in Table 1, while those of other European patients already reported in literature can be found in Appendix A.

### 2.1. Family A

The proband (II.1) is a European male who was born from unrelated healthy parents (Figure 1A).

At 18 months of age, the patient developed left facio-brachial weakness associated with focal motor seizures. Brain magnetic resonance imaging (MRI) with MR angiography (MRA) showed severe bilateral narrowing of the terminal portions of the ICAs and several bilateral strokes, especially in the distal parietal and temporo-occipital areas, due to coexistent stenosis of the right posterior cerebral artery (PCA) with insufficient collateral flow (Figure 2A–D).

Dynamic susceptibility contrast (DSC) enhanced perfusion imaging revealed severe reduction of the signal in the cerebral blood flow (CBF) map, especially in MCA territories. Digital subtraction angiography (DSA) confirmed the suspected diagnosis of severe bilateral moyamoya. Acetylsalicylic acid (ASA) therapy was then started, and the patient underwent indirect revascularization surgery by one-stage bilateral multiple burr holes technique (MBHT). On the first postoperative day, bilateral and extensive MCA strokes occurred (Figure 2E), so a right decompressive craniotomy was performed. During hospitalization in the intensive care unit and while undergoing treatment with intravenous administration of midazolam and phenobarbital to obtain seizure control and antibiotics (cefazoline and vancomycin), the child presented elevated aminotransferases (ALT, 1200 IU/L; AST, 900 IU/L) with increased levels of gamma-glutamyl transferases (GGT, 600 IU/L) and serum bile acid (100 mg/dL). Abdomen ultrasound showed abnormality of liver echotexture evoking steatohepatitis and abdominal computed tomography angiography (CTA) revealed hypoplasia of the distal abdominal aorta. Given the presence of cholestasis, the patient was treated with ursodeoxycholic acid during the acute phase, resulting in a progressive normalization of liver enzymes. No other major or minor causes of liver disease were found during follow-up. Over time, the child’s liver enzymes became almost completely normalized. Stenosis of the left pulmonary artery was detected by echocardiography. At two-year follow-up, spastic tetra paresis (treated by intrathecal baclofen therapy) and severe motor and language delay recurred, with a Pediatric Stroke Outcome Measure of 9/10 [20].

### 2.2. Family B

The proband (III.2) is a European girl, the second child of non-consanguineous parents (Figure 1B). Her older brother (III.1) and father (II.1) were healthy. The mother (II.2) reported a long history of headache without aura. The maternal grandfather (I.1) was affected by blood pressure hypertension (BPH), chronic lower limb venous insufficiency, diffuse atherosclerotic disease, and myxomatous mitral valve disease.

For the proband, growth and psychomotor milestones had been normal and no major health problems were reported until age 7. Because of persistent headache and seizures characterized by brief motor arrest and loss of consciousness, she underwent brain MRI and MRA. Thinning of distal ICAs, complete signal loss of MCAs, A1 segments of ACAs, and the left PCA, as well as stenosis of the P2 segment of the right PCA were detected by MRA and subsequently confirmed by DSA; various areas of signal reduction in MCA and ACA territories, especially in the left hemisphere, were detected on the CBF map generated from DSC perfusion imaging (Figure 2F–H). Electroencephalography (EEG) documented left temporo-occipital slowing during hyperpnea. Valproic acid (20 mg/kg/day) and ASA (100 mg/day) therapy were started. Homocysteine, IgM and IgG anticardiolipin, and anti-beta 2 Gp1 were normal, as were coagulation, factor V Leiden, and lupus anticoagulant screen ratio. The patient underwent bilateral indirect revascularization using MBHT (Figure 2I).

After a five-year follow-up, the headaches disappeared. She remained seizure-free with normal video EEG patterns, and anti-seizure medication was subsequently withdrawn.

No strokes occurred during the eight-year follow-up. MRA evidenced collateral circles in the basal deep perforating arteries and from activated cortico-dural connections between arteries arising from branches of the external carotid arteries and leptomeningeal network (Figure 2J). Liver function and blood pressure were always normal.

A diffuse and mild (max 9.7 mm at TSA level) wall thickening of all arteries, likely due to intimal thickening, was detected at ultrasound of the supra-aortic trunks and abdominal aorta.

### 2.3. Family C

The proband (III.3) is a European male, the second child of non-consanguineous parents (Figure 1C). The older brother (III.1) was referred as healthy. The patient’s mother (II.2) underwent surgery for atrial myxoma. She had BPH and showed livedo reticularis (LR) (Figure 3A). She also exhibited short stature.

At 5 years of age, the proband presented with acute onset of right-side hemiparesis and dysarthria. MRI and MRA showed severe stenosis of ICAs and bilateral signal loss of A1 and M1 segments, and an acute stroke in the left frontal region in the anterior MCA territory; posterior hypertrophic perforating vessels were also detected, confirmed by conventional cerebral angiography; signs of bilateral hypoperfusion of anterior districts were detected on brain perfusion images (Figure 2K–N). ASA therapy was started and bilateral one-stage indirect revascularization using MBHT was performed.

The child also presented persistent BPH, partially controlled by calcium channel blockers. Echo-color-Doppler of heart and large supra-aortic and subdiaphragmatic arteries was normal. He also had recurrent transient LR triggered by cold exposure. An extensive rheumatologic work-up resulted normal. During his follow-up, the patient developed moderately elevated aminotransferases (ALT, 520 IU/L; AST, 300 IU/L) and high levels of GGT (260 IU/L) triggered by an Epstein–Barr virus infection. Due to the persistence of cholestasis beyond intercurrent infection, an extensive diagnostic evaluation was conducted to exclude any other major infectious, autoimmune, neoplastic, and metabolic causes of liver disease. Given the biochemical evidence of persistent cholestasis, choleretic treatment with ursodeoxycholic acid was administered, with prompt normalization of liver enzymes. A rapid increase in transaminase levels was observed after suspension of therapy, which rapidly resolved after restoring the full dosage. Magnetic resonance cholangiopancreatography (MRCP) and abdominal CTA were normal. After one year, a specular cortical ischemia appeared in the right frontal region (Figure 2O). During a three-year follow-up, right hemiparesis and dysarthria significantly improved; an improvement in cerebral perfusion was also highlighted in the DSC-MR study.

The patient recently developed flat annular rashes with central clearing on the lower limbs (Figure 3A).

### 2.4. Molecular Diagnosis

To investigate the genetic cause of MMA in the three affected children, deep exome sequencing was performed for each proband and their parents (family trio). A potentially causative missense variant of *RNF213* was identified in all probands in heterozygous state. Segregation analysis suggested a role for these variants in the pathogenesis of MMA in index cases as well as in minor vasculopathies of other carrier relatives, in accordance with an autosomal dominant inheritance and incomplete penetrance (Table 2).

In family A, the proband showed a *de novo* missense variant in exon 46 of *RNF213* (c.12358T>C; p.Phe4120Leu), not previously reported (Appendix A). This amino acid change was classified as pathogenic based on different in silico predictive algorithms (Table 2). Interestingly, a different nucleotide substitution (c.12360C>G) causing the same amino acid change was very recently reported and associated with severe MMA occurring before 3 years of age [21].

In family B, the proband also presented a heterozygous missense variant in exon 46 of *RNF213* (c.12353C>G; p.Ser4118Cys), predicted to be likely pathogenic (Table 2; Appendix A). This variant was inherited from the proband’s mother, and the maternal grandfather was also identified as a carrier. The grandfather subsequently underwent brain MRI and intracranial MRA, showing severe stenosis of the supraclinoid segment of the right ICA, the ipsilateral proximal segment of the MCA and of the ACA, and proximal stenosis of the left MCA. Occlusion of the PCAs and thin leptomeningeal collaterals were also present (Figure 4A–F). The amino acid change falls very close to that detected in the sporadic case of family A (Figure 5). A different nucleotide substitution at the same position (c.12535C>T) that changes serine 4118 to phenylalanine (p.Ser4118Phe) was previously described, occurring *de novo* in a 3-month-old female with seizures, arterial narrowing involving the internal carotid and intracranial arteries and the inferior abdominal aorta, as well as persistently elevated transaminases [22].

In the proband of family C, we identified a heterozygous missense variant (c.14599G>A; p.Glu4867Lys) in exon 62 of *RNF213* (Appendix A). This variant, not previously reported, was inherited from her mother, in whom it occurred *de novo*. Different in silico predictive algorithms classified this variant as of uncertain significance (Table 2).

For each proband, no other causative variants were detected in genes possibly associated with MMA (Appendix A).

The aminoacidic substitutions observed in our cases involved residues highly conserved among species (Figure 5). These missense variants and those previously identified in European MMA patients are all clustered in the C-terminal region of RNF213 protein, mainly in the E3 ligase domain (Figure 6 and Appendix A).

The ubiquitin ligase activity of RNF213 is thought to depend on the E3-RING domain, where many of these variants are localized [23]. The amino acid changes identified in families A and B do not substantially modify the chemical properties of the residues involved, while the amino acid change in family C replaces an acid residue with a basic residue. Three-dimensional (3D) homology modeling did not show any evidence of structural damage for the p.Phe4120Leu and p.Ser4118Cys variants (Families A and B; Figure 7), for which both the literature and in silico prediction tools provide clear evidence of pathogenicity (Table 2) [21,22]. Conversely, for the novel p.Glu4867Lys variant identified in family C and classified in silico as of uncertain significance, 3D homology modeling showed the loss of the hydrogen bond between the side chain of Glu^4867^ and the backbone of Leu^4870^ (Figure 7) and a predicted expansion of cavity volume by 121.824 Å^3^ [24].

## 3. Discussion

Since its first definition and characterization in 1957 [25], a growing body of literature has been produced on the pathogenesis, clinical features, and neurosurgical treatment of MMA. Although the pathogenetic mechanisms underlying MMA are yet to be completely clarified, it soon became clear that a genetic cause is highly likely for several reasons. MMA can be associated with several genetic conditions, including autosomal dominant, recessive, and X-linked disorders, as well as with chromosomal aberrations, which has further complicated the dissection of its genetic basis. Despite being rare, familial cases of MMA have also been reported.

The high prevalence of MMA in East Asians compared to other populations prompted the scientific community to look for a genetic susceptibility factor, subsequently identified as the p.Arg4810Lys variant of *RNF213* in East Asian populations [8]. In recent years, an increasing number of genetic disorders and genetic susceptibility factors leading to moyamoya have been identified, mainly thanks to the large amount of molecular data made available by the massive use of NGS analysis [26,27].

*RNF213*, also known as Mysterin, encodes for a multi-domain protein of 591 kDa, the largest E3 ubiquitin ligase in the human proteome. Its 3D structure was recently determined in mouse and consists of an N-terminal stalk, a dynein-like core with six ATPase units, and a multidomain E3 module (Figure 7) [23]. In the last few years, increasing evidence has suggested an important role for RNF213 in lipid metabolism, as well as in cellular response to hypoxia [28,29]. Further, RNF213 seems to be connected to the ubiquitin-proteasome system, changes which are caused by *RNF213* knock-down [28,29,30]. Specifically, RNF213 is able to activate NF-kB signaling, possibly influencing angiogenesis via the expression of inflammatory cytokines [28,31].

Rare variants in *RNF213* have also been identified as a susceptibility factor in MMA patients of non-Asian ancestry [13]. In Europeans, rare missense variants that significantly predispose to MMA are clustered in a C-terminal hotspot of *RNF213* (Figure 6). The need to identify additional factors among *RNF213* variants is suggested by the fact that only 25% of mutation carrier relatives are affected by MMA [16]. Furthermore, *RNF213* variants have been associated with non-MMA vasculopathies [32], while other genes have been linked to MMA, in some cases characterized by syndromic phenotypes [26,27,33].

Here, we report the cases of three children with severe bilateral MMA caused by rare missense variants in *RNF213* clustered in the C-terminal region, in line with literature data.

We also describe the incomplete penetrance of MMA traits and the co-occurrence of other clinical and neuroradiological findings in the patients and their carrier relatives, supporting previous observations correlating *RNF213* variants to syndromic forms of MMA [22]. Specifically, amino acid substitutions at positions 4118–4120 and close to position 4867 seem to be associated with a syndromic condition characterized by typical cerebrovascular MMA, susceptibility to hypertransaminasemia, and stenosis of extracerebral arteries [22].

In family A, the affected patient (II.1; Figure 1A) presented the p.Phe4120Leu substitution in *RNF213*, occurring *de novo* and affecting a highly conserved amino acid (Figure 5; Table 2). The same amino acid change, caused by a different nucleotide substitution, was very recently reported in a male of European ancestry with bilateral MMA surgically treated at 5 years and hypertension due to a right renal artery occlusion discovered at 8 years, and who subsequently further required aorto-left renal artery bypass at 23 years [21]. Harel et al. previously reported the very close p.Ser4118Phe substitution in *RNF213* in a 3-month-old female with seizures, arterial narrowing involving the ICA, intracranial arteries, and inferior abdominal aorta, and persistent hypertransaminasemia (Appendix A) [22]. The phenotype of our patient perfectly recapitulates the phenotype reported by Harel et al. Interestingly, both these amino acid changes fall in a highly conserved region close to the RING domain (Figure 5). As already suggested [21,22], these rare variants could lead to severe and very early onset MMA, with occlusion of other arteries including the abdominal aorta, and renal, iliac, and femoral arteries. Our results support this hypothesis and could contribute to delineating a new syndromic form of moyamoya, characterized by the triad of moyamoya, inferior abdominal aorta stenosis, and liver susceptibility with recurrent elevated aminotransferases. The transient liver involvement in patients with *RNF213* variants, possibly triggered by stressful events (e.g., viral infections and hypoxic states), may be linked to endothelial cell dysfunction, as suggested by a mouse model lacking *Rnf213*, in which abnormal post-ischemic angiogenesis seems to be amplified in critical conditions [34]. Further studies should investigate whether elevated aminotransferases might also involve patients carrying *RNF213* variants without MMA and, more in general, whether other *RNF213* variants may be related to liver disease.

Serine 4118 is changed to cysteine in the affected patient in family B (III.2; Figure 1B, Figure 5). This rare and not previously reported *RNF213* variant was inherited from the maternal grandfather. In the grandfather, stenosis of MCAs, the distal right ICA, and ipsilateral ACA was assessed with MRA. To our knowledge, none of the affected and carriers from family B had elevated aminotransferases.

The patient M003_4 reported by Guey et al. harbored the substitution p.Asp4122Val (Appendix A) [16]. This variant is very close to the variants p.Phe4120Leu and p.Ser4118Cys found in our probands from families A and B, respectively. Interestingly, this patient also presented severe pulmonary arterial hypertension and had a hepatitis at 14 months, unsolved at time of report.

In terms of extracerebral involvement, stenosis of the left pulmonary artery and hypoplasia of the distal abdominal aorta were present in the patient from family A, while the patient from family B exhibited intimal thickening of the inferior abdominal aorta. Other subjects in all the three families investigated presented BPH. The homozygous p.Arg4810Lys missense variant in *RNF213* was recently reported to be associated with intracranial atherosclerosis and systemic vasculopathy (e.g., peripheral pulmonary artery stenosis and renal artery stenosis) [35].

The wall thickening highlighted by ultrasound in patient III.2 from family B involved all major explorable arteries. This represents a novel finding. Similarly, Pinard et al. described the first patient in whom the femoral arteries were small in caliber [21]. Intimal thickening with fibrosis and damaged vascular smooth muscle cells are distinguishing features of MMA. Similar stenotic phenomena responsible for MMA are likely to involve extracerebral vessels theoretically able to cause complete stenosis.

The novel missense variant at position 4867 of *RNF213* seems to be associated with skin findings, blood hypertension, and variable cerebrovascular manifestations.

In the affected patient of family C (III.3; Figure 1C), the identified p.Glu4867Lys substitution is novel and was inherited from his mother, in whom it occurred *de novo*. They were both affected by BPH and LR (Figure 3A,B), and the mother had an atrial myxoma. She recently underwent skin biopsy following the appearance of annular figurate erythema of the buttocks 6 months previously (Figure 3B). Histopathological findings showing parakeratosis, spongiosis, and dense perivascular lymphocytic infiltration suggested a diagnosis of erythema annulare centrifigum (EAC) (Figure 3C). Intriguingly, two years later, the child developed the same clinical features involving both legs (Figure 3A). We did not have the opportunity to examine the boy’s grandfather to determine whether he had had similar skin manifestations. Skin involvement was representative in family C and included two major clinical presentations, LR and EAC. LR is a well-known skin condition indicating reticulate erythema usually affecting extremities. It may be either physiological (newborns), caused by environmental/climate changes, or associated with impairment of blood flow. Although non-specific, LR has been associated with MMA [36,37]. EAC is a figurate erythema, often presenting with annular figures, and is considered a non-specific hypersensitivity reaction to a given antigen whose etiology remains unknown. It may present as superficial, with a typical inner border of desquamation, or as a deep variant. Some of the many causes reported in literature are infections, food or drugs, cancer, and systemic diseases [38]. No previous studies describe an association between EAC and MMA. We suggest including MMA in the growing list of possible causes of this peculiar cutaneous eruption. Strong et al. very recently reported the cases of two children with severe MMA and congenital liver, kidney, and skin disease [39]. The authors identified two *de novo* and previously unreported heterozygous missense variants in *RNF213* (p.Leu4139Trp and p.Cys4856Arg) and described similar skin findings consisting in multiple, flat annular rashes with central clearing. Skin biopsy was non-specific for the patient with the p.Leu4139Trp substitution but suggestive of erythema multiforme in the patient with the p.Cys4856Arg mutation. Interestingly, the position of this latter amino acid change is very close to the one we observed in family C. A livedo racemosa was observed in three out of five families (M010, M039, and M261) reported by Grangeon et al. [17]. Among these families, only M010 carried a *RNF213* variant (Appendix A). Two patients belonging to the M039 and M261 families and presenting livedo racemosa also had a dysimmune thyroiditis. Our patients with skin abnormalities had normal thyroid hormones and sierology.

Different in silico prediction tools classified the p.Glu4867Lys substitution from family C as a variant of uncertain significance, although missense variants with limited evidence of pathogenicity have already been reported and associated with MMA (Appendix A). This amino acid change involves a highly conserved residue (Figure 5) and falls within the E3 ligase domain of RNF213. In support of a possible functional effect, changes in chemical property and protein conformation were suggested by 3D modeling of wild-type and mutated RNF213 protein (Figure 7). The ubiquitin ligase activity of RNF213 is probably due to the RING domain, around which the majority of *RNF213* variants associated with MMA reported to date are clustered. However, a small subgroup of potentially pathogenic variants, including p.Glu4867Lys, are localized distally to the RING domain in the E3-core domain (Figure 6). Together, these observations strongly support the pathogenicity of the *RNF213* missense variants identified in the families reported here.

The age of onset and the heterozygous or homozygous status are worth discussing. There is some evidence that clinical presentation in MMA patients from East Asia carrying the p.Arg4810Lys substitution is dosage dependent. The homozygous state for this allele was in fact associated with early-onset MMA (before 5 years) with severe symptoms at diagnosis, mainly due to cerebral infarction, and poor neurocognitive long-term outcome [40]. Homozygous patients also showed a high penetrance rate of systemic vascular involvement presenting a very unique pattern of diffuse narrowing of the aorta and iliofemoral arteries, together with stenosis of renal, celiac, or peripheral pulmonary arteries, regardless of the presence or absence of MMA [40]. Heterozygous patients for the p.Arg4810Lys substitution were mostly asymptomatic or had isolated MMA [32]. In terms of extracerebral involvement, MMA seems to represent only the tip of the iceberg. As recently suggested by Bang et al., extracranial involvement in MMA patients includes coronary, aorta, and iliofemoral arteries, together with stenosis of renal, celiac, or peripheral pulmonary arteries [32].

Here we provide further evidence that the heterozygous status for some rare missense variants in *RNF213* is sufficient to cause early onset of MMA and extracerebral artery stenosis/narrowing in Europeans. The segregation study combining molecular and neuroradiological data also revealed cerebrovascular disease in patient I.1 from family B, who presented bilateral MMA with occlusion of the PCAs and thin leptomeningeal collaterals. Unlike in childhood-onset MMA, typical angiographic features might not be observed in the early phase of adult-onset MMA [41,42].

Our report, together with findings from the available literature, supports the causative role of some rare *RNF213* variants for MMA in Europeans as in East Asians. In addition, these variants can lead to distinctive syndromic forms of MMA, with a positional effect of some of these mutations. For example, variants affecting positions 4118–4122 seem to be specifically linked to subdiaphragmatic aortic stenosis or pulmonary arteries abnormality and the chance to observe elevated aminotransferases, while more C-terminal variants to skin findings. Further studies are required to confirm this hypothesis and to explore the underlying pathogenetic mechanisms.

Our findings and several other papers on MMA in Europeans suggest that a comprehensive molecular testing should be offered to all the Europeans affected with MMA (i.e., NGS panel or whole exome sequencing), considering that genes different to *RNF213* could also be involved. This can help clinicians in recognizing a genetic cause when present, looking for other specific and gene-based manifestations, making a proper genetic counseling, and estimating the recurrence risk.

A model of autosomal dominant inheritance and incomplete penetrance has been proposed for familial MMA [1,43]. *RNF213* falls in this scheme, then suggesting molecular screening of first-degree relatives. Similarly, the growing number of familial cases of MMA in Europeans suggests that it may be worth to extend investigations to the carrier relatives. A combined genetic and clinic-radiologic approach may be useful. It remains to be understood in which cases to do it and at what age it could be surely informative for carrier relatives of MMA patients with variants in *RNF213* or in other genes.

## 4. Materials and Methods

### 4.1. Patient Recruitment

Probands were affected by bilateral MMA and followed at the Pediatric Units of the University of Campania “Luigi Vanvitelli” and the Santobono-Pausilipon Children’s Hospital. Written informed consent for blood sample collection and genetic investigation was obtained from all the probands and their relatives involved in the study, according to the Declaration of Helsinki. For each subject, genomic DNA or RNA was extracted using standard procedures, when necessary.

### 4.2. Exome Sequencing

For each proband and their parents (family trio), deep exome sequencing was performed using Agilent SureSelect Clinical Research Exome (family A) or Agilent SureSelectXT Custom Constitutional Panel (families B and C), according to the manufacturer’s instructions (Agilent Technologies, Santa Clara, CA, USA).

Sequencing was performed using the NextSeq 500 or Novaseq 6000 system (Illumina, San Diego, CA, USA). The mean coverage of targeted regions was 98.3% at 10×, ensuring the detection of genetic variants with high sensitivity and specificity. Sequence reads were mapped to the reference human genome assembly (February 2009, GRCh37/hg19) and analyzed by the BWA enrichment pipeline and a second independent in-house pipeline [44,45]. Calling of single nucleotide variants (SNVs) and small insertions/deletions (Ins/Del) was performed with the Genome Analysis Toolkit (GATK) (gatk.broadinstitute.org). Called SNVs and Ins/Del variants were annotated using ANNOVAR.

For data filtering, we considered: (1) variants that passed quality control and with more than 10 reads; (2) variants with a frequency <1% in global and European populations as well as in the Genome Aggregation Database (gnomad.broadinstitute.org); (3) variants that were not reported in our internal database of 3747 exomes; (4) variants occurring *de novo* and standard models of Mendelian inheritance; (5) variants with a potential effect on gene function and predicted to be pathogenic/likely pathogenic (SIFT, Polyphen, MutationTaster, PROVEAN, ClinVar). Candidate variants were classified in accordance with ACMG guidelines [46]. According to these criteria, no further variants were selected among MMA genes reported in Appendix A. These genes were selected from the cerebral vascular malformations (Version 2.59) panel (https://panelapp.genomicsengland.co.uk/panels/147/, accessed on 2 March 2020), filtered for “moyamoya disease” and enriched with three recently suggested MMA genes [17,27,47].

### 4.3. Variant Validation and Segregation Analysis

*RNF213* candidate variants were annotated according to HGVS nomenclature on RefSeq NM_001256071.3 and validated by segregation analysis after PCR amplification of the specific exons of *RNF213* and their flanking regions in the probands, their parents, and other relatives. Primer pairs used are available on request. PCR products were double strand sequenced using BigDye Terminator sequencing chemistry (Life Technologies, Carlsbad, CA, USA) and analyzed on an ABI 3130xL automatic DNA sequencer (Life Technologies Carlsbad, CA, USA).

### 4.4. Bioinformatic Tools

Different in silico analysis tools were used to classify variants and investigate the possible pathogenic effect of amino acid changes identified in *RNF213*. ACMG classification [46] was established using Varsome (varsome.com) and segregation data. Pathogenicity was evaluated using different algorithms, such as SIFT and PROVEAN (provean.jcvi.org/index.php) [48,49], PolyPhen-2 (genetics.bwh.harvard.edu/pph2/) [50], Mutation Taster (www.mutationtaster.org) [51], CADD (cadd.gs.washington.edu/snv) [52], and MutPred2 (mutpred2.mutdb.org) [53]. 3D homology modeling of human RNF213 protein was based on the recently published cryo-EM structure of mouse RNF213 (RCSB-PDB: 6TAX) [23] and generated using Phyre2 (swissmodel.expasy.org) [54]. Missense3D (missense3d.bc.ic.ac.uk/~missense3d) was used to investigate any possible effect of the identified missense variants on the 3D structure [24]. Alignment of RNF213 protein sequences from *Homo sapiens* (NP_001243000.2), *Mus musculus* (NP_001035094.2), *Rattus norvegicus* (XP_038943469.1), *Cricetulus griseus* (XP_035293728.1), *Gorilla gorilla* (XP_018882703.2), *Bos taurus* (XP_002696190.2), and *Ovis aries* (XP_012041858.2) was obtained using Clustal Omega [55] at UniProt (https://www.uniprot.org/).

### 4.5. Literature Search Strategy

By using PubMed and EMBASE databases, we searched for previously reported European MMA patients with *RNF213* pathogenetic variants, collecting their clinical and radiological data. The terms “RNF213”, “moyamoya”, and “European” or “Caucasian” were searched in title or abstract of all English scientific literature available. We also checked references of all the studies on European patients with MMA and *RNF213* to rescue eventually lost papers.

## 5. Conclusions

Our report, together with previous literature data, confirms that rare variants in *RNF213* can cause MMA and other cerebrovascular anomalies in Europeans as in East Asian populations, with a pattern of incomplete penetrance and variable expressivity. *RNF213* may also represent the major genetic causative factor of MMA among Europeans. Further, our data possibly delineate a novel syndromic form of MMA linked to mutations affecting positions 4118–4120 and 4867 characterized by elevated aminotransferases and extracerebral vascular involvement. The same triad plus a peculiar skin involvement (mainly LR and/or EAC) might be related to substitutions of more C-terminal amino acids distally to the RING domain in the E3-core domain.

In children with MMA and variants in *RNF213*, it is necessary to perform a comprehensive assessment of extracerebral arteries, achieve optimal blood pressure control to improve prognosis, look for other systemic involvement, and carry out a thorough analysis of carrier relatives.

## Figures and Tables

**Figure 1 ijms-23-08952-f001:**
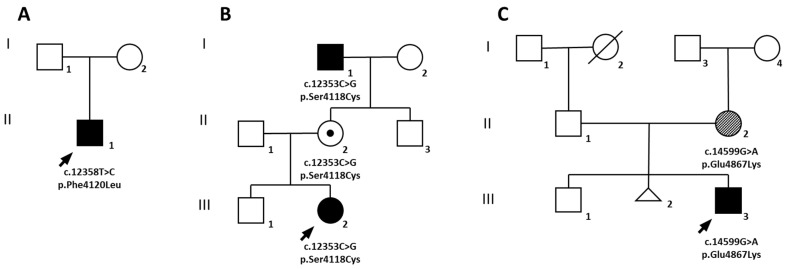
*Pedigree of families **A**, **B** and **C** with variants in RNF213.* Probands are indicated by an arrow. A black symbol identifies subjects with bilateral moyamoya, a dot indicates an obligate carrier, while a diagonal pattern fill highlights other vascular manifestations. Generations are reported in roman numbers.

**Figure 2 ijms-23-08952-f002:**
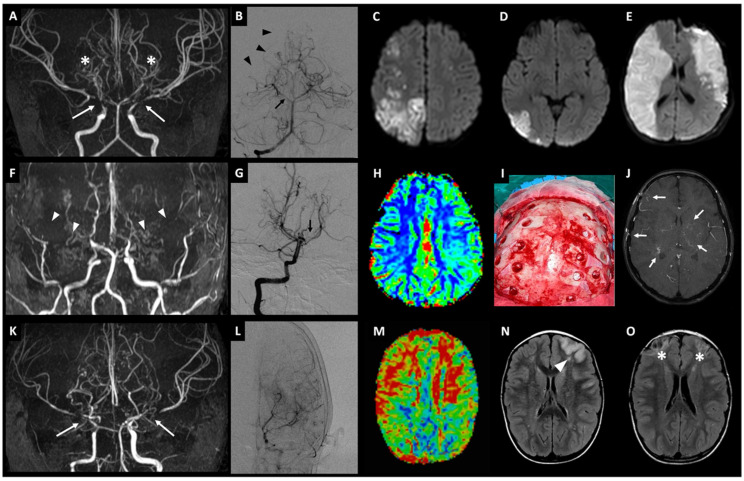
*Imaging from probands of families A-C with bilateral moyamoya.* Family A, patient II.1 (**A**–**E**): Magnetic resonance angiography (MRA) image showing severe bilateral narrowing of internal carotid arteries (ICAs; white arrows) and typical hypertrophic perforating collateral vessels (“puff of smoke” appearance; white asterisks) (**A**); vertebral digital subtraction angiogram showing stenosis of the right posterior cerebral artery (PCA; black arrow) and a less extensive pial network (black arrowheads) (**B**); diffusion-weighted MR images showing acute ischemic strokes in parietal and temporo-occipital areas at onset (**C**,**D**) and subsequent massive bilateral strokes in middle cerebral artery districts on the first postoperative day (**E**). Family B, patient III.2 (**F**–**J**): MRA image showing severe bilateral narrowing of internal carotid arteries with moyamoya collateral vessels and lack of both proximal anterior and middle cerebral arteries (white arrowheads) (**F**); vertebral angiogram showing stenosis of the left PCA (black arrow) (**G**); cerebral blood flow (CBF) map of MR perfusion study showing impaired cerebral perfusion, especially in the left hemisphere (**H**); intraoperative image of indirect revascularization obtained by one-stage bilateral multiple burr holes technique (**I**); follow-up MRA image showing hypertrophic collateral network at the basal ganglia and along the postoperative dural surface (white arrows) (**J**). Family C, patient III.3 (**K**–**O**): MRA images showing typical moyamoya pattern with severe bilateral stenosis of the terminal ICA, A1, and M1 segments (white arrows), and hypertrophic deep vessels (**K**), confirmed on angiography (**L**); mean transit time (MTT) map of MR perfusion study showing prolongation of MTT due to impaired perfusion in bilateral anterior districts (**M**); MR fluid-attenuated inversion recovery (FLAIR) images showing acute ischemic left frontal stroke at onset (white arrowhead) (**N**) and delayed bilateral frontal ischemic lesions after one year (white asterisks) (**O**).

**Figure 3 ijms-23-08952-f003:**
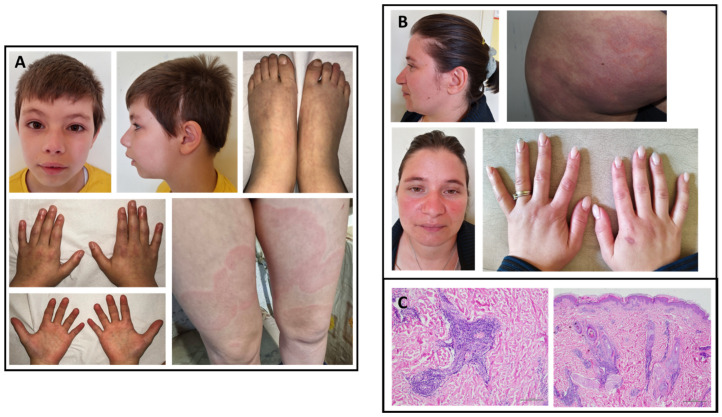
*Clinical pictures and histological images of proband and his carrier mother from family C.* Proband (III.3; family C) shows some facial peculiarities such as arched eyebrows, mild retrognathia, scalp scar due to neurosurgery, livedo reticularis of hands, feet, and legs, and skin lesions compatible with annular figurate erythema involving both upper legs (**A**). Proband’s mother (II.2) shows no dysmorphic features but presents livedo reticularis and annular figurate erythema (**B**). Histological images of skin biopsy samples taken from the mother (hematoxylin and eosin staining; 4× and 10× magnification), showing a dense perivascular lymphocytic infiltrate involving both superficial and deep dermal vascular plexus. Epidermis was normal (**C**).

**Figure 4 ijms-23-08952-f004:**
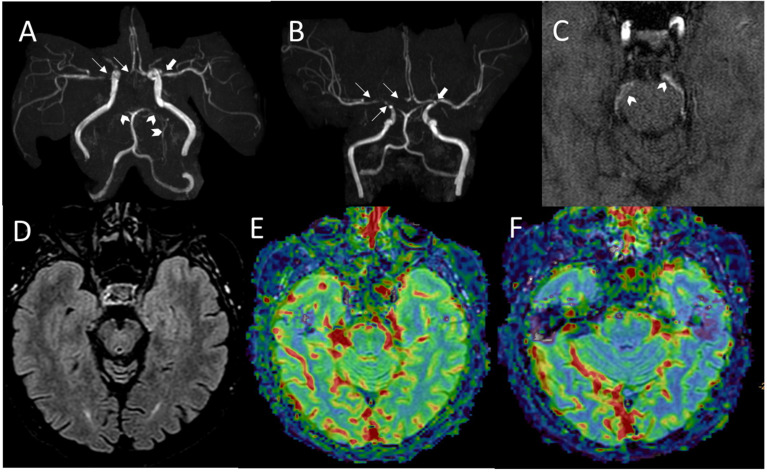
*Imaging of maternal grandfather (I.1) from family B*. Family B, patient I.1 (**A**–**F**): MRA images (**A**–**C**) showing severe stenosis of both the right ICA supraclinoid segment and the ipsilateral M1 segment, with signal loss of the right A1 segment (thin arrows in (**A**,**B**)), with mild proximal stenosis of the left M1 segment (thick arrows in (**A**,**B**)). Flow-related signal loss in distal opercular and sylvian branches is also visible bilaterally. Occlusion of PCAs with development of perimesencephalic thin collaterals are also evident (arrowheads in (**A**–**C**)). FLAIR axial image at the level of temporo-occipital lobes and midbrain showing normal tissue signal (**D**). CBF and cerebral blood volume maps generated from dynamic susceptibility contrast-enhanced perfusion weighted imaging showing leptomeningeal and cortical signal reduction in the left temporo-occipital regions (**E**,**F**).

**Figure 5 ijms-23-08952-f005:**
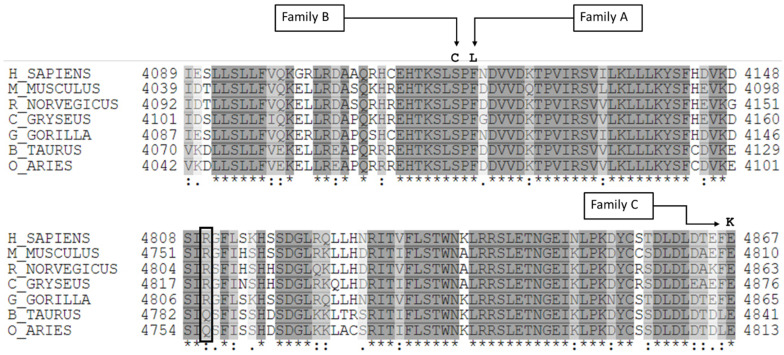
*Alignment of RNF213 protein from different species.* Amino acid changes identified in our patients all affected highly conserved residues among species (*Homo sapiens*, *Mus musculus*, *Rattus norvegicus*, *Cricetulus griceus*, *Gorilla gorilla*, *Bos taurus*, and *Ovis aries*). A box indicates the position of Arg^4810^, the canonical polymorphic variant in East Asian populations; (*) positions with a single, fully conserved residue; (:) positions with conservation between amino acids with similar properties; (.) positions with conservation between amino acids of weakly similar properties.

**Figure 6 ijms-23-08952-f006:**
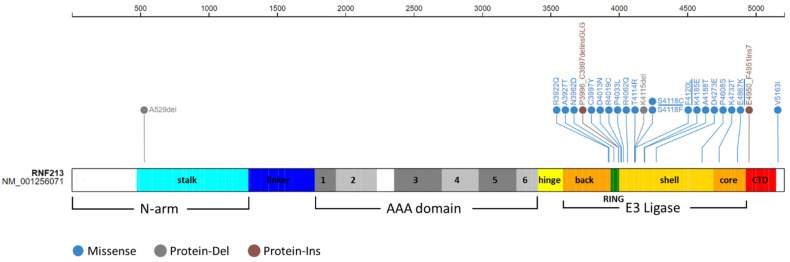
*Graphical view of RNF213 protein showing its functional domains and published pathogenic variants reported in European populations*. Functional motifs of RNF213 (5207 amino acids) are differently colored. Pathogenic variants associated with moyamoya angiopathy are color grouped according to their functional effect. The variants identified in the present study are underlined.

**Figure 7 ijms-23-08952-f007:**
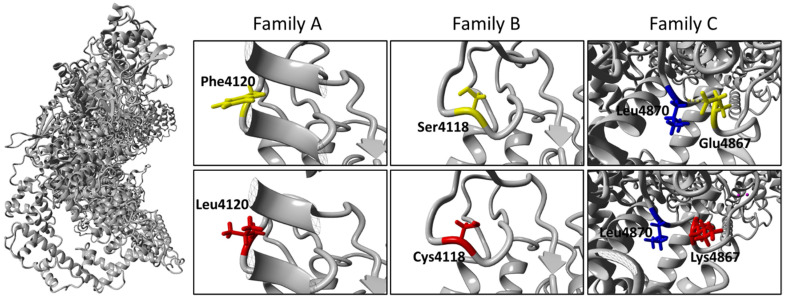
*3D homology modeling.* 3D homology modeling of human RNF213 protein (NP_001243000) based on the cryo-EM structure of mouse RNF213 (RCSB-PDB: 6TAX) for wild-type and mutant forms. Magnification of residues affected by substitution are shown for each family, with the side chain of wild-type residue (yellow) on the upper panel, and the side chain of mutated residue (red) on the lower panel. The hydrogen bond is lost between the side chain of Leu^4870^ (blue) and Glu^4867^.

**Table 1 ijms-23-08952-t001:** Demographical, clinical, and radiological features of our patients.

Patient ID	Family A	Family B			Family C III.3	
	II.1 (Index)	III.2 (Index)	II.2	I.1	III.3 (Index)	II.2
**Sex**	M	F	F	M	M	F
**Mutation**	p.Phe4120Leu	p.Ser4118Cys			p.Glu4867Lys	
**MMA/other cerebrovascular condition/bilateral or unilateral**	MMA/B	MMA/B	No	MMA/B	MMA/B	No
**Age at onset (y)**	1.5	7	n.a.	n.r.	5	n.r.
**MMA symptoms**	Left facio-brachial weakness associated with focal motor seizures	Persistent headache and seizures characterized by brief motor arrest and loss of consciousness	n.a.	None	Acute onset of right-sided hemiparesis and dysarthria	n.a.
**Atypical angiographic features**	Posterior cerebral arteries P1 and P2 tracts	n.r.	n.r.	Occlusion of the PCAs and thin leptomeningeal collaterals	n.r.	n.r.
**HBP**	No	No	No	Yes	Yes	Yes
**Extracerebral arterial or cardiac anomalies**	Pulmonary artery L branch stenosis, hypoplasia of distal abdominal aorta	Generalized thickening of arterial walls	None	Chronic lower limb venous insufficiency, diffuse atherosclerotic disease, myxomatous mitral valve disease	None	Atrial myxoma
**Hypertransaminasemia/hepatitis**	Transitory hypertransaminasemia	No	No	No	Elevated transaminases triggered by Epstein-Barr virus infection	No
**Other systemic features**	None	None	None	None	LR, delayed refill, EAC	LR, EAC (confirmed by pathology), facial fibromata, short stature

Abbreviations: ID = patient identifier; B = bilateral; F = female; M = male; HBP = high blood pressure; MMA = moyamoya angiopathy; LR = livedo reticularis; PCA = posterior cerebral artery; EAC = erythema annulare centrifigum; n.a. = not applicable; n.r. = not reported.

**Table 2 ijms-23-08952-t002:** Variants identified in RNF213. Segregation analysis and in silico evaluation of their pathogenicity.

Patient	Family A	Family B			Family C	
	II.1 (Index)	III.2 (Index)	II.2	I.1	III.3 (Index)	II.2
Age (yrs)	4	14	45	68	8	40
Gender	M	F	F	M	M	F
Genomic (Hg19)	chr17-78343600-T-C	chr17-78343595-C-G			chr17-78360109-G-A	
cDNA (NM_001256071.3)	c.12358T>C	c.12353C>G			c.14599G>A	
Exon	46 of 68	46 of 68			62 of 68	
Protein	p.Phe4120Leu	p.Ser4118Cys			p.Glu4867Lys	
Inheritance	*de novo*	Maternal	Paternal	n.d.	Maternal	*de novo*
ACMG Classification	Pathogenic (PM2;PM1;PS1;PS2)	Likely pathogenic (PM1;PM2;PM5)			Uncertain significance (PM2;BP4)	
MAF (gnomAD)	0	0			0	
SIFT	Deleterious (score: 0)	Deleterious (score: 0)			Tolerated (0.11)	
Polyphen-2	Probably damaging (score: 0.976)	Probably damaging (score: 1.000)			Possibly damaging (score: 0.493)	
PROVEAN	Deleterious (score: −5.433)	Deleterious (score: −4.450)			Neutral (score: −2.200)	
MutationTaster	Disease causing (score: 0.98)	Disease causing (score: 0.99)			Polymorphism (score: 0.86)	
CADD	26.1	25.9			21.7	
MUtPred2 (cutoff = 0.5)	0.697	0.735			0.607	

Abbreviations: M = male; F = female; n.d. = not determined.

## Data Availability

The data presented in this study are available on request from the corresponding author.

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
