# Peer review of "Mystery(n) Phenotypic Presentation in Europeans: Report of Three Further Novel Missense RNF213 Variants Leading to Severe Syndromic Forms of Moyamoya Angiopathy and Literature Review"

_ijms, 2022, doi:10.3390/ijms23168952_

Round 1

Reviewer 1 Report

Moyamoya angiopathy (MMA) is a rare cerebral vasculopathy, and RNF213 is a major susceptibility gene for MMA. In this study, the authors found three rare variants of RNF213 (p.Phe4120Leu, p.Ser4118Cys, and p.Glu4867Lys,) in three European families including MMA probands by exome sequencing. Genetical and clinical characteristics of these families were analyzed especially focusing on extracerebral vascular anomalies and clinical phenotypes other than vasculopathy. The authors evaluated the found RNF213 variants using in silico prediction and 3D modeling tools.

Many previous reports have identified rare RNF213 variants in European patients with MMA, so novelty of this study is limited in this regard. On the other hand, the analysis in this paper focusing on the syndromic form of MMA is informative and interesting. However, the data presentation and discussion of this point is unorganized and insufficient, and needs to be improved

Comments

1. The literature search and extraction methods should be described in a manner like systematic review studies. I think that some applicable studies are not included in Table 1.

2.  Table 1 is very difficult to read because it is long and not well designed in layout. Furthermore, most of it is unrelated to the main purpose of the paper and is not mentioned in the text. The table in the main text should be revised to show only the focused cases. I recommend that the whole table be shown as a supplemental material after the improvement of its layout design.

3. Table 1 shows that both “Proband” of the Ref. 21 and “Patient 1” of the Ref. 16 have RNF213 p.Ser4118Phe. However, only the case of the Ref. 21 was discussed in main text. Are these two cases the same? If so, please state it. If not, “Patient 1” of the Ref. 16 should also be mentioned in the discussion part.

4. “M003_4” of the Ref 16 in Table 1 harbor p.Asp4122Val. This variant is very close to the found variant: p.Phe4120Leu and p.Ser4118Cys. The author should discuss this case.

5. There seems to be a discrepancy between the information in Table 1 and Figure 1. Please check the following points

1) I.1 of Family B is “MMA/B” in Table 1, but “other cerebrovascular abnormalities” in Figure 1

2) II.2 of Family B is ”No" MMA and other cerebrovascular condition in Table 1, but “other cerebrovascular abnormalities” in Figure 1

3) II.2 of Family C is "No" MMA and other cerebrovascular condition in Table 1, but “other cerebrovascular abnormalities” in Figure 1

6. Parts of Legends of Fig2 and 3 were inserted into the main text. I think that the authors have not adequately checked the manuscript. Please ask a professional proofreader to check your manuscript.

7. Page 7; “For each proband, no other causative variants were detected in genes possibly associated with MMA.” Please show the gene names of “genes possibly associated with MMA” and provide a list of found variants in these genes as supplemental material.

8. Page 9-10: “They were both affected by BPH and LR (Figure 3A,B), and the mother had an atrial myxoma. She recently underwent skin biopsy following the appearance of annular figurateerythema of the buttocks 6 months previously (Figure 3B). Histopathological findings showing parakeratosis, spongiosis, and dense perivascular lymphocytic infiltration suggested a diagnosis of erythema annulare centrifigum (EAC) (Figure 3C). Intriguingly, two year later the child developed the same clinical features involving both legs (Figure 3A).”

This part should be described in the result section.

9. Please show sanger sequencing charts of the found three variants, and provide primer information using sequencing.

10. Please show a high resolution image of Figures 2 and 3.

11. Please check Ref. 33. This study did not show abnormal post-ischemic angiogenesis in a mouse model lacking Rnf213.

Reviewer 2 Report

Introduction and abstract: - “Moyamoya angiopathy (MMA) is a rare cerebral vasculopathy most commonly occurring in children.” I am not sure that this sentence is correct. Epidemiological data is sparse, espeicially in Europe, please delete “…most commonly occurring in children”.

- “Though MMA is being increasingly reported among Europeans, very little is cur[1]rently known about the penetrance, mode of inheritance, and clinical phenotype of pa[1]tients with MMA and rare variants in RNF213”, please also clarify that in Caucasians there is no founder mutation as in East Asians: Genomewide association study identifies no major founder variant in Caucasian moyamoya disease. Liu W, Genet. 2013 Dec;92(3):605-9

Results:

- What is the term which the journal´s editor prefer? European Caucasian, Caucasian or European? I prefer European Caucasian, but perhaps there is a norm within the journal?

- Unfortunately, the quality of MRAs and angiograms in my version is low, I am not sure if vasculopathy really represent MMA or a Moyamoya-like angiopathy? I would ask the authors to send me (markus.kraemer1977@gmxde) some high definition files to be sure that the term MMA is really correct or have to be changed to Moyamoya-like angiopathy. Figure 2A resembles straight running arteries like in ACTA 2 mutation, moreover, the journal should depict the figures in a higher size

Discussion:

Page 10/29: “Although non-specific, LR has been associated with MMA”, please add one or two discussion sentences concerning other skin features in MMA based on Mitri et al. Eur J Neuro 2021 May;28(5):1784-1793, please note that in Grangeon´s paper also livedo racemosa was described (supplemental and discussion)

- Discussion: please add two or three sentences concerning usefulness of genetic screening in Caucasian MMA patients, is it advisable now in clinical practice or are further studies needed? Does incomplete penetrance argue against systematic screening of first degree relatives? Despite missing AMORE study results, perhaps antiplatelets are useful in early detected but still asymptomatic relatives?

- How do you explain incompete penetrance? Is there another co-mutation as PALD! As in Grangeon´s paper?

-

Reviewer 3 Report

The authors reported the three rare missense variants, except for the p.Arg4810Lys variant of RNF213 in three children with MMD in Europe by using the whole exome sequencing. Indeed, the results seem to be descriptive not conclusive in terms of the definition for causative mutation of RNF213. However, the analysis and discussion of their data as well as literature review are interesting.

 At the moment, the most important thing is to collect the data of mutation of RNF213 with MMD as much as possible, since we have still not clarified the relation between the biomechanical dysfunction of RNF213 associated with its mutation and MMD.

Round 2

Reviewer 1 Report

Thank you for your response. I believe the manuscript has been appropriately revised.